# Parent Reports of Developmental Service Utilization After Newborn Screening

**DOI:** 10.3390/ijns11010003

**Published:** 2024-12-31

**Authors:** Elizabeth Reynolds, Sarah Nelson Potter, Samantha Scott, Donald B. Bailey

**Affiliations:** RTI International, 3040 E. Cornwallis Road, Research Triangle Park, P.O. Box 12194, Research Triangle Park, NC 27709, USA; snpotter@rti.org (S.N.P.); sjscott@rti.org (S.S.); dbailey@rti.org (D.B.B.)

**Keywords:** newborn screening, early intervention, developmental services, developmental delays, severe combined immunodeficiencies, congenital hypothyroidism, Pompe disease

## Abstract

Newborn screening (NBS) presents an opportunity to identify a subset of babies at birth who are at risk for developmental delays and could benefit from a range of developmental services. Potential developmental services in the United States include Part C Early Intervention (EI), private therapies, and school-based services. Using parent-reported outcomes, this study examined the rates at which a sample of children diagnosed with NBS conditions used each developmental service. An online survey of 153 parents representing children with 27 different NBS conditions found that nearly 75% of children (*n* = 112) used at least one developmental service, with private therapies being the most frequent. Children were referred to EI relatively early and were often eligible because their medical diagnosis automatically qualified them. When examining condition-specific results for children with severe combined immunodeficiencies, congenital hypothyroidism, and Pompe disease, we found variability in rates of use, with high rates overall. Our findings suggest that many children diagnosed with an NBS condition continue to have developmental delays even after they receive appropriate medical care. Future research with more systematic follow-up is needed to understand whether the NBS program facilitates entry into these services and whether more streamlined processes could benefit children and families.

## 1. Introduction

Newborn screening (NBS) is a public health program that screens nearly all infants within the first few days of life for a panel of rare medical conditions, including metabolic, endocrine, hematological, and other genetic disorders. A substantial number of these babies are at risk for developmental delays and could benefit from a range of developmental intervention services [1,2,3,4]. Developmental services in the United States include state-based Part C Early Intervention (EI) programs, private therapies (e.g., physical therapy at a private practice), and school-based services (e.g., special education).

The extent to which children diagnosed as a result of NBS use developmental services is unknown due to the absence of a systematic way of collecting and aggregating these data [5,6,7,8]. This gap hinders our ability to target specific populations, provide information and guidance to families as they seek services, and inform providers about the specific needs for children with NBS conditions. Drawing on previous research based on parent-reported outcomes [9,10], this study begins to address this gap with a sample of parents whose children were diagnosed with an NBS disorder. We present preliminary results on developmental service use among children and highlight the need for expanded and systematic NBS long-term follow-up on children’s developmental outcomes.

### 1.1. Newborn Screening Conditions and Developmental Delay

Although NBS has improved morbidity and mortality outcomes [11], it is known that not all symptoms for many disorders are resolved after early detection and initial medical treatment [1,11]. Even after treatment, children may be at a heightened risk for experiencing intellectual disability, hearing and vision problems, heighted anxiety and depression, attention deficits, and behavioral problems [2,3,12,13,14]. One recent study documented risk of developmental delay for the dried blood spot conditions on the Recommended Uniform Screening Panel (RUSP) in the United States [1]. Of the 34 conditions examined, most (*n* = 29) conditions put children at a moderate to high risk of developmental delay and/or high risk of medical complexity, with cascading negative developmental outcomes, even after early detection and treatment [15].

### 1.2. Developmental Services

Three primary avenues exist for children in the United States to access developmental services (Table 1). EI is the state-based system of services provided to children and their families from birth to age 3 through Part C of the Individuals with Disabilities Education Act (IDEA) [16]. Children must be referred and deemed eligible to receive EI services. To be eligible, children must have a documented developmental delay or have an Established Condition that has a “high probability of resulting in a developmental delay” [16]. There is considerable variability in the degree to which NBS conditions are included on states’ Established Conditions lists [1,17]. Once a child is determined to be eligible for EI, they are provided tailored services based upon their Individualized Family Service Plan (IFSP). Services may include developmental monitoring or specific therapies such as physical, speech, or occupational therapy; nutritional care; and psychological support. Some states provide EI services for families for free, whereas others use a combination of federal funds, private insurance, Medicaid, and/or fees paid by the family [18].

Private therapies typically include similar services to EI but do not have any age restrictions and are not administered, funded, or regulated by states. For example, private therapy services may include speech–language therapy (SLT) or physical therapy (PT) at a private clinic. Families may have out-of-pocket costs depending on whether services are covered by insurance and, if so, provided within their insurance network.

Children older than 36 months of age who qualify for special education services have an Individualized Education Plan (IEP) describing goals and services (e.g., special education and related services) in school through Part B of IDEA [23]. Children are eligible for an IEP if they have a disability and demonstrate need for special education services through evaluation. IEPs describe the specific education services a child needs at school.

Overall, these services differ by ages of children they support, referral and eligibility requirements, location of services, types of services, and funding. Yet they each support children to achieve developmentally appropriate skills, prevent loss of function, and can mitigate the consequences or complications associated with medical diagnoses [24].

### 1.3. Present Study

Using data from an online survey of parents representing children with a range of NBS conditions, the current study addresses three research questions: (1) At what frequency are children being referred to, deemed eligible for, and enrolled in EI? (2) At what frequency are children using private therapy services? (3) At what frequency do children have education supports (i.e., an IEP)?

Following analyses for the above research questions, we conducted exploratory analyses to understand developmental service rates for three specific conditions: severe combined immunodeficiencies (SCID), congenital hypothyroidism (CH), and Pompe disease. We selected these conditions because they vary in the extent they are associated with developmental delays and medical complexity (Table 2), and we had survey responses from 10 or more parents for each condition.

## 2. Materials and Methods

### 2.1. Participants

Data for the current study were derived from an online survey of parents. Inclusion criteria included the following: (1) respondents were parents or legal guardians of children who had an NBS disorder, (2) children were identified via NBS and formally diagnosed prior to their first birthday, and (3) children primarily lived in the United States for their first 3 years of life. We used a descriptive survey design with convenience sampling. Overall, data from 153 parents were included in this study.

### 2.2. Survey Instrument

Our research team developed a novel survey intended for parents of children who were diagnosed with an NBS condition. Following development of our initial survey draft, we conducted qualitative pretest interviews with three parents [38]. The purpose of these interviews was to ensure survey clarity and identify any questions that were difficult to answer. Based on their feedback and comments, the survey was revised.

The final survey was published in Alchemer, an online survey platform. The survey included 49 items and was expected to take about 10 min to complete. Nearly all survey questions were multiple-choice or checkbox responses. The survey was available in English only.

### 2.3. Procedures

Several strategies were used to recruit study participants. When available, we contacted disease-specific patient advocacy groups (PAGs) and requested they share our survey link. If disease-specific PAGs were not available or unresponsive, we contacted disease-specific online support groups or admins of private Facebook groups. Additionally, we communicated directly with parents who had publicly available websites describing their journey with their child with the medical condition, as well as NBS Ambassadors [39]. If receptive, we asked them to share our survey link with others in their community. Lastly, we reached out to US-based specialty clinics and state NBS follow-up centers to share our survey with specific patient populations. Data were collected from parents between October 2022 and May 2023, and participants were offered a USD 10 gift card for survey completion.

The survey data were analyzed using SAS Studio, version 9.04. Descriptive data analyses included means, median, and percentages.

## 3. Results

### 3.1. Results from the Full Sample (N = 153)

Demographic Information

On average, children were 5.7 years old at time of survey completion (median: 3 years old; range: 1 month to 31 years; Table 3). The average age at diagnosis was 0.4 months (median: 0 months; range: 0 months to 6 months). There was geographic diversity in our respondents, with 39 out of 51 states (including the District of Columbia) represented. Regions represented included the South (35.5%; *n* = 54), Midwest (25.7%; *n* = 39), Northeast (23.0%; *n* = 35), and West (15.8%; *n* = 24). Most children were White (86.9%; *n* = 133) and not Hispanic or Latino (91.5%; *n* = 140).

A total of 27 unique disorders were represented in our survey results, including 3 disorders not on the RUSP at the time the survey was completed (Table 4). The disorders most represented included CH (12.4%, *n* = 19), Pompe disease (11.8%; *n* = 18), and mucopolysaccharidosis type I (8.5%, *n* = 13).

### 3.2. Overall Service Utilization Rates

Of the 153 parents, 112 (73.2%) reported their child used at least one developmental service, and 12.4% of children (*n* = 19) used all three developmental services. An additional 9.8% (*n* = 15) of children used both EI and private therapies, and 9.2% (*n* = 14) of children used both private therapies and school-based services. Lastly, 11.1% (*n* = 17) used EI services only, 24.8% (*n* = 38) used private therapies only, and 1.3% (*n* = 2) used school-based services only. Of the 41 children who did not use any developmental service, 23 were under the age of 3 and therefore not yet eligible to receive school-based services.

#### 3.2.1. Early Intervention 

EI Referral: Nearly half of parents (45.8%; *n* = 70) reported their child was referred for EI. For those referred children, specifics on the age of referral and person who made the referral can be found in Table 5.

EI Eligibility: Parents of 61 children (39.9%) said their children were ultimately eligible for EI. For those eligible, parents could select multiple reasons for eligibility, and their responses can be found in Table 5.

#### 3.2.2. Private Therapies

More than half (56.2%, *n* = 86) of parents reported that their child received at least one type of private therapy. For those who received private therapies, the top three most frequently used private therapies are listed in Table 6.

#### 3.2.3. School-Based Services

There were 86 children in our sample who were age 3 years or older (Table 7). The mean and median age of this subsample (*n =* 86) was 9.3 years and 8 years, respectively. Of these, nearly half (47.7%; *n* = 41) of parents reported their child had an IEP.

#### 3.2.4. Exploratory Analyses of Exemplar Conditions

Severe Combined Immunodeficiencies (*n* = 10; Table 8): Overall, 5 out of 10 (50%) children with SCID used at least one developmental service. No child with SCID used all three services. Two children (20%) used both EI and private therapies. One (10%) child used EI only, and two (20%) used private therapies only.

Of the 10 children with SCID, half (50%; *n* = 5) were referred to EI prior to their first birthday. Referrals came from PCPs (40%; *n* = 2), specialist physicians (40%; *n* = 2), or state NBS program and/or follow-up staff (20%; *n* = 1). Of the five children with SCID who were referred to EI, three (60%) were eligible for EI. Reasons for eligibility included developmental delay, being at risk for developmental delay, and/or having an automatically qualifying medical condition. The three most utilized EI services included PT, OT, and surveillance.

Four out of ten (40%) children with SCID used private therapies. The most utilized private therapies included PT and OT.

Most (80%; *n* = 8) children with SCID were under 3 years of age and were therefore not old enough to be eligible for school-based services. Neither of the two children older than 3 years of age (mean age: 7 years) reported having an IEP.

Congenital hypothyroidism (*n* = 19; Table 8): Overall, 14 out of 19 (73.7%) children with CH used at least one developmental service. Two (10.5%) children with CH used all three services. Three (15.8%) children used both EI and private therapies. Three (15.8%) children used both private therapies and school services. One (5.3%) child used EI only, four (21.1%) used private therapies only, and one (5.3%) used school services only.

Ten of the nineteen (52.6%) children with CH were referred to EI. One parent whose child was not referred said they believed their child should have been referred to EI. Of those referred, most (80%; *n* = 8) were referred prior to their first birthday. Referrals were made by either their specialist physician (30%; *n* = 3) or PCP (30%; *n* = 3), NBS and/or follow-up staff (30%; *n* = 3), or self-referral (10%; *n* = 1). Of those 10 who were referred, most (60%; *n* = 6) were eligible for EI. Reasons for eligibility included developmental delay, being at risk for developmental delay, and/or having an automatically qualifying medical condition. All six of the children with CH who were eligible for EI went on to use EI services. The three most utilized EI services included SLT, PT, and OT.

Twelve out of nineteen children (63%) with CH used at least one private therapy. The most utilized private therapies included SLT, hearing/audiology, OT, and vision.

Twelve children (63.2%) with CH were aged 3 years or older (mean age: 7.3). Of these children, half (50%; *n* = 6) had an IEP at school.

Pompe disease (*n* = 18; Table 8): Overall, 14 out of 18 (77.8%) children with Pompe disease used at least one developmental service. Two (11.1%) children with Pompe used all three services. Four (22.2%) children used both EI and private therapies. None used both private therapies and school services. Four (22.2%) children used EI only, four (22.2%) used private therapies only, and none used school services only.

Eleven out of eighteen (61.1%) children with Pompe were referred to EI. Two (11.1%) parents whose children were not referred said they believed their children should have been referred to EI. Of those eleven who were referred, eight (72.7%) were referred prior to their first birthday. Referrals were made by their specialist physician (45.5%; *n* = 5), by self-referral (36.4%; *n* = 4) or by their PCP (18.2%; *n* = 2). Of those 11 who were referred, 10 (90.9%) were ultimately eligible. Reasons for eligibility included having an automatically qualifying medical condition (80%; *n* = 8) or a developmental delay (40%; *n* = 4) and/or being deemed at risk for developmental delay (30%; *n* =3). All 10 eligible children utilized services. The top three EI services included PT, OT, and SLT.

Ten out of eighteen children (55.6%) with Pompe disease used at least one private therapy. The top private therapies included PT, feeding, SLT, assistive technology, nursing, and parent education.

Only four (22.2%) children with Pompe disease were 3 years of age or older (mean age: 4.3 years). Of these children, two (50%) had an IEP.

## 4. Discussion

The present study provides preliminary results of the developmental service utilization rates of children identified through NBS. Our study demonstrates how parent-reported outcomes can generate initial data and findings but also highlights the need for more systematic data collection as part of NBS follow-up. Indeed, we hope this type of small-scale study helps justify a comprehensive national approach to data collection. In addition to summarizing our findings below, we highlight specific areas of need for future research.

Nearly 75% of children (*n* = 112) used either EI, private therapies, and/or school-based services. In our condition-specific analyses, we found variability between conditions in rates of use. For example, 50% of children with SCID, 74% of children with CH, and 78% of children with Pompe used at least one developmental service. Our study corroborates previous research and demonstrates that even after early detection and prompt medical treatment, children diagnosed with NBS conditions may require and use specific services because of developmental delay [1].

EI is intended for infants and young children under the age of 3 years. Earlier entry maximizes benefits to children and families before they need to transition out of the program, although it is sometimes a challenge for EI programs to identify and refer appropriate infants [40,41]. The current study is the first to examine the age at which children with NBS conditions are referred to EI. We found that more than 80% (*n* = 57) of referred babies were referred prior to their first birthday, and 50% (*n* = 34) were referred prior to 6 months of age. This finding is important because other studies suggest that the average age of referral for EI in the United States is closer to 14 months [42,43]. An IDEA coordinator survey found that the median age of referral was 18 months [44]. Our results suggest that an NBS diagnosis may prompt and/or facilitate earlier referral.

The processes by which children are referred to EI after a positive NBS diagnosis are not clear. Previous studies suggest that referrals directly from NBS staff are infrequent, though there is variability across states, programs, and conditions [15]. The results from the current study corroborate these previous findings. When reporting who referred their child to their state’s EI program, “state NBS and/or follow-up staff” was the option *least* frequently endorsed by parents, after “specialist physician,” “primary care provider,” and “self-referral.” Nonetheless, previous studies suggest that referrals directly from NBS could be beneficial to expedite entry into EI [15], reduce burden on PCPs and specialist physicians [45,46], and alleviate barriers to accessing EI programs for underrepresented children [4,15,47,48].

In the United States, children are eligible for EI if they have a medical condition that has a “high probability of resulting in a developmental delay” [49]. Federal guidelines provide broad categories of such conditions, including chromosomal abnormalities, genetic or congenital disorders, and inborn errors of metabolism [50]. In contrast to this broad regulatory language, some states have created “Established Conditions lists” that name specific medical diagnoses that are automatically eligible. However, NBS conditions are infrequently included on these lists [1]. Despite this, results from the current study demonstrate that having a “medical condition that automatically qualified child” was the *most* frequently endorsed reason children were eligible for EI. The reason most children in our survey were automatically eligible was because of their medical diagnosis, despite the low incidence of NBS conditions on Established Condition lists. This generates important questions: are children diagnosed with NBS automatically eligible for EI because their condition fits into one of the broad categories (e.g., inborn errors of metabolism)? Or are they automatically eligible because their condition is explicitly included on an “Established Conditions list”? Which (broad category versus Established Condition list) is more streamlined for families and clinicians?

Overall, the condition-specific findings for utilization of EI show variability. For example, 38% (*n* = 58) of all children in our sample ultimately received EI. Rates of EI utilization were 50% (*n* = 3) for children with SCID, 32% (*n* = 6) for children with CH, and 55.6% (*n* = 10) for children with Pompe. Despite significant differences in the clinical presentation of these conditions and some variation in utilization rates, a case could be made to view all NBS conditions as a single broad category when considering eligibility for EI [1]. Although there is no federal definition for what constitutes a “high probability of developmental delay,” nearly 40% of children in our sample using EI indicate relatively high rates of delay or suspected delay. Similarly to the existing broad categories in the federal eligibility descriptions (e.g., inborn errors of metabolism), “newborn screening conditions” could be a broad category triggering automatic eligibility. Indeed, one state (Michigan) already specifies that “all disorders tested for in the Michigan Newborn Screening Program” are Established Conditions and deem affected children automatically eligible for EI services [51].

Compared to rates of EI (38%, *n =* 58), rates of utilization of private therapy were higher (56%; *n* = 86) among children diagnosed with an NBS condition. This may be a result of the relatively strict eligibility and requirements for EI (see Table 1) or the age at which the child is referred to services. Additionally, this difference could be a result of parent decision to use private therapies in lieu of or in addition to EI services.


*Limitations and Future Directions*


There are limitations to our study that are important to note. First, while our findings suggest that many children diagnosed as a result of NBS ultimately have developmental delays and could benefit from developmental services, data to understand long-term developmental outcomes for most of the NBS conditions do not exist. Our study relied on parent-reported outcomes. Consistent assessments and improved data collection efforts for the long-term follow-up of children diagnosed as a result of NBS are warranted.

Relatedly, our sample was a convenience sample and was not a randomly selected cohort representing all NBS conditions. Despite our efforts to include at least one response from each RUSP disorder, not every condition was represented. Our convenience sample may not be representative of the entire population of affected individuals, especially in terms of race, ethnicity, and socioeconomic status. Conducting future studies with more systematic sampling is important because service utilization may differ based on race, ethnicity, and/or socioeconomic status [52,53,54]. It is also possible that we had higher engagement with developmental services because our sample was mostly White and relatively well connected in their disease space given our recruitment methods. These families were potentially also more connected to the developmental service programs available to them.

Another important limitation to our study was the relatively young age of the participants. Only 56% (*n* = 86) of children were older than age 3 and therefore eligible for school-based services. And of those older than age 3, the median age was 8 years old. Indeed, an older sample may show that a higher percentage of children use school-based services, further highlighting the need for longer-term data tracking, as well as systems of data coordination between education and health departments.

Finally, our study does not examine the effectiveness of these types of developmental services. There is some evidence that EI has additive benefits for children with complex medical diagnoses [55,56]. Relatedly, our study does not examine the additive benefit of multiple types of services; it is unknown, for example, whether accessing both EI and private therapies is beneficial. Overall, a systematic, long-term examination of these outcomes is necessary given the value in understanding the benefits of NBS beyond survival for children and families affected by these rare and complex medical conditions [57].

## 5. Conclusions

The results from our parent survey demonstrate high utilization rates of developmental services after a diagnosis via NBS. This indicates that despite medical treatment, many children diagnosed with an NBS disorder experience developmental delay. Our results show that children were referred to EI relatively early and were eligible for EI because their medical diagnosis automatically qualified them. However, specific questions remain about how NBS facilitates entry into these services and whether our results are generalizable to a broader sample of children with disorders identified via NBS. This study lays important groundwork for future studies to examine whether streamlined processes, such as automatically qualifying all NBS conditions for EI, may be warranted to benefit children and families.

## Figures and Tables

**Table 1 IJNS-11-00003-t001:** Developmental services in the United States.

Criteria	Early Intervention	Private Therapies	School-Based Services
Definitions given to survey participants	State-based program for children birth to 3 years of age. Response to federal legislation to ensure early support and services to infants and babies with developmental delays and disabilities; services available because of the special education law (Individuals with Disabilities Education Act [IDEA]) [19]. Types of services are determined by each child’s Individualized Family Service Plan (IFSP). Example: children might receive monthly care coordination appointments, speech–language therapy (SLT) in their home, or physical therapy (PT) in their daycare. To read about your states’ Early Intervention (EI) program, click here: https://www.cdc.gov/ncbddd/actearly/parents/state-text.html	Includes any additional or supplementary services a child receives outside of state-based EI programs. Services for children at any age. Might include services in a private practice, at the hospital, or through medically complex children’s waivers. - Example: children might receive SLT at a private clinic, PT at a hospital, or care coordination through medically complex children’s waivers.	N/A
Federal regulations	Part C	N/A	Part B
Age	Birth to 3 years of age	No age restrictions	3 to 21 years of age
Referral process	Anyone can make a referral, including caregivers. Parental consent is required before evaluation or starting services.	Anyone can make a referral. Depends on insurance: referral from physician may be required.	Anyone can make a referral, including caregivers. Parental consent is required before evaluation or starting services.
Eligibility requirements	Eligibility is set by Part C of IDEA [16]. Children must have a documented developmental delay (established by comprehensive multidisciplinary assessment) or an Established Condition that has a “high probability of resulting in a developmental delay”; some states include “at risk for developmental delay” as an eligibility category. States vary in required documentation of developmental delay. Established Condition lists vary by state.	Eligibility will vary based on private therapy’s enrollment criteria. Insurance determines necessary documentation of reason for treatment (e.g., speech therapist may conduct specific evaluations and use required standardized measures to determine need).	Eligibility is set by Part B of IDEA [20,21]. A child must be determined to have a disability and need special education services through an evaluation.
Location of services	At the child’s home or other “natural environment” (e.g., daycare).	Flexibility in service setting varies by provider/program (e.g., private practice clinic, community clinic, hospital setting, child’s home, telehealth).	School in the “least restrictive environment.”
Types of services	“System of services” for child and family: - Screening; - Evaluations and assessments for IFSP developmental monitoring; - Coordination of EI services (e.g., PT, occupational therapy [OT], SLT, medical services); - Transition (including to Part B/Exiting Part C).	Variety of services: - SLT; - OT; - Feeding/nutrition services; - PT; - Behavioral health; - Psychological services; - Evaluations/assessments; - Counseling.	Individualized Education Plans describe the specific special education including instructions, supports, and services a child needs at school; can include specific therapies such as PT or SLT.
Funding	Some services are free for families, including coordination, evaluation, and development of the IFSP. Variety of sources. Grants from federal government. State supplemental funding. Some states bill child’s insurance and/or Medicaid for services listed on the IFSP.	Often billed to child’s insurance. Families may seek out-of-network providers, requiring them to cover out-of-pocket costs. Some providers may not take insurance, requiring families to pay out of pocket.	Schools are required to provide free appropriate public education to eligible students with disabilities. Variety of sources: - Grants from the federal government; - State supplemental funding.

Source: Bailey, 2021 [22].

**Table 2 IJNS-11-00003-t002:** Potential of developmental delay for three example newborn screening conditions.

Condition	Description	Potential of Developmental Delay
Severe combined immunodeficiencies (SCID)	Group of primary immune deficiency diseases. Usually fatal in the first year of life without treatment [25,26,27]. Early symptoms include infection, diarrhea, and failure to thrive. Treatment frequently involves a hematopoietic stem cell transplantation (HSCT).	HSCT has been shown to improve survivability of SCID, but treatment protocols may negatively affect children’s developmental outcomes [25,28]. Clinical recommendations include developmental assessments and monitoring, and prompt referral for supportive interventions [26]. Recommended to be automatically qualified for EI based on high risk of medical complexity [1].
Congenital hypothyroidism	An endocrine condition that affects thyroid hormone levels [29]. When untreated, leads to intellectual disability. Treatment includes hormone replacement medication.	Even after early detection and treatment, children are at risk for emotional–behavioral problems [30], hearing loss [31], and developmental delay (DD) [1]. Care guidelines recommend monitoring for delays, speech problems, attention and/or memory problems, and behavioral problems [31,32] and prompt referral for developmental services as necessary [31]. Recommended to be automatically qualified for EI based on moderate risk of DD in treatment-altered natural history [1].
Pompe disease	A lysosomal storage disorder; includes infantile and late-onset phenotype [33]. Symptoms include hypotonia and hypertrophic cardiomyopathy in infancy. Treatment includes enzyme replacement therapy [33].	Even after treatment, Pompe disease puts children at risk for muscle weakness, speech and language disorders, cognitive delays, and learning problems [34,35,36]. Published guidelines for the treatment of Pompe disease include recommendations for developmental services, such as physical therapy, occupational therapy, speech–language therapy, and EI [37]. Recommended to be automatically qualified for EI based on high risk of DD in treatment-altered natural history and high risk of medical complexity [1].

**Table 3 IJNS-11-00003-t003:** Demographic data (*N* = 153).

Demographic	Data	
**Age in years** (Mean, *SD*)	5.7	5.9
**Geographical Region** (%, *n*)		
South	36.0	55
Midwest	25.5	39
Northeast	22.9	35
West	15.6	24
**Race** (%, *n*)		
White	86.9	133
Black or African American	10.5	16
Other	2.6	4
**Ethnicity** (%, *n*)		
Hispanic	8.5	13
Non-Hispanic	91.5	140

Note: **South** includes Alabama, Arkansas, Delaware, District of Columbia, Florida, Georgia, Kentucky, Louisiana, Maryland, Mississippi, North Carolina, Oklahoma, South Carolina, Tennessee, Texas, Virginia, and West Virginia. Midwest includes Illinois, Indiana, Iowa, Kansas, Michigan, Minnesota, Missouri, Nebraska, North Dakota, Ohio, South Dakota, and Wisconsin. Northeast includes Connecticut, Maine, Massachusetts, New Hampshire, New Jersey, New York, Pennsylvania, Rhode Island, and Vermont. West includes Alaska, Arizona, California, Colorado, Hawaii, Idaho, Montana, Nevada, New Mexico, Oregon, Utah, Washington, and Wyoming.

**Table 4 IJNS-11-00003-t004:** Diagnoses for full sample (*N* = 153).

Diagnosis	%	*n*
**Primary congenital hypothyroidism**	**12.4**	**19**
**Glycogen storage disease type II (Pompe)**	**11.8**	**18**
Mucopolysaccharidosis type I	8.5	13
Congenital adrenal hyperplasia	7.8	12
Medium-chain acyl-CoA dehydrogenase deficiency	7.2	11
Cystic fibrosis	6.5	10
**Severe combined immunodeficiencies**	**6.5**	**10**
S,S disease (Sickle cell anemia)	5.2	8
3-methylcrotonyl-CoA carboxylase deficiency	3.9	6
Galactosemia	3.3	5
Krabbe disease *	3.3	5
Maple syrup urine disease	3.3	5
Propionic acidemia	3.3	5
Glutaric acidemia type I	2.6	4
Long-chain L-3 hydroxyacyl-CoA dehydrogenase deficiency	2.0	3
Methylmalonic acidemia (cobalamin disorders)	2.0	3
Phenylketonuria	2.0	3
Hearing loss	1.3	2
Isovaleric acidemia	1.3	2
X-linked adrenoleukodystrophy	1.3	2
Alpha thalassemia *	0.7	1
Carnitine palmitoylatransferase 2 deficiency *	0.7	1
Homocystinuria	0.7	1
Sickle cell disease	0.7	1
Spinal muscular atrophy	0.7	1
Trifunctional protein deficiency	0.7	1
Very long-chain acyl-CoA dehydrogenase deficiency	0.7	1

Note: Example diagnoses are in bold. Disorders marked with an asterisk (*) were not on the Recommend Uniform Screening Panel in the United States at the time of survey completion. Krabbe disease was added to the RUSP in July 2024.

**Table 5 IJNS-11-00003-t005:** Frequency of children diagnosed with a condition via newborn screening being referred to, deemed eligible for, and enrolled in Early Intervention.

Early Intervention (EI) Referral, Eligibility, and Enrollment	%	*n*
**EI Referral**		
Referred to state’s EI program (*N* = 153)	45.8	70
Age of child when they were referred (*n* = 70)		
Before 12 months	81.4	57
13–24 months	11.4	8
15–36 months	7.1	5
Who referred child (*n* = 70)		
Primary care provider	22.9	16
Self-referral	17.1	12
Specialist physician	37.1	26
State newborn screening and/or follow-up staff	14.3	10
Other/I don’t know	8.6	6
Children were not referred, but parents believe they should have been (*n* = 76)	14.5	11
**EI Eligibility**		
Determined to be eligible for EI services (*N* = 153)	39.9	61
Reasons for eligibility; could select multiple reasons (*n* = 61)		
Medical condition automatically qualified child	65.6	40
Documented developmental delay	44.3	27
At risk for developmental delays	29.5	18
Other/I don’t know	4.9	3
**EI Services Received**		
Received state’s EI services (*N* = 153)	37.9	58
Top 3 most frequently used EI services *(n* = 58)		
Physical therapy	60.3	35
Speech–language therapy	58.6	34
Occupational therapy	55.2	32

EI Services Received: Of the 61 children who were referred to and eligible for EI, 58 received EI services, indicating that 37.9% of the full sample ultimately received EI services. For those who received services, the top 3 most frequently used EI services are listed in Table 5.

**Table 6 IJNS-11-00003-t006:** Frequency of children diagnosed with a condition via newborn screening using private therapy services.

Private Therapies (*N* = 153)	%	*n*
Used at least one private therapy (*N* = 153)	56.2	86
Top 3 most frequently used private therapies (*n* = 86)		
Speech–language therapy	39.5	34
Physical therapy	31.4	27
Nutrition services	29.1	25

**Table 7 IJNS-11-00003-t007:** Frequency of children diagnosed with a condition via newborn screening having educational supports (Individualized Education Plans).

School-Based Services (*N* = 86)	%	*n*
Had an Individualized Education Plan (*N* = 86)	47.7	41

**Table 8 IJNS-11-00003-t008:** Service utilization rates for children with SCID (*n* = 10), CH (*n* = 19), and Pompe disease (*n* = 18) and comparison to the full sample (*N* = 153).

	N	Average Age	Used at Least One Service (EI, Private, IEP) n (%)	EI n (%)	Private Therapies n (%)	n	Average Age	IEP n (%)
						Limited to sample of children > age 3 yrs		
Exemplar conditions								
SCID	*N* = 10	2.5 years	5 (50%)	3 (30.0%)	4 (40.0%)	*n* = 2	7 years	0 (0%)
CH	*N* = 19	5.1 years	14 (73.7%)	6 (31.6%)	12 (63.0%)	*n* = 12	7.3 years	6 (50.0%)
Pompe	*N* = 18	1.7 years	14 (77.8%)	10 (55.6%)	10 (56.0%)	*n* = 4	4.3 years	2 (50.0%)
Full sample	*N* = 153	5.7 years	112 (73.2%)	58 (37.9%)	86 (56.2%)	*n =* 86	9.3 years	41 (47.7%)

Note. CH = congenital hypothyroidism; EI = Early Intervention; IEP = Individualized Education Plan; NBS = newborn screening; SCID = severe combined immunodeficiencies.

## Data Availability

Deidentified datasets generated during and/or analyzed during the current study are available from the corresponding author on reasonable request.

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
