# Peer review of "Parent Reports of Developmental Service Utilization After Newborn Screening"

_2409-515X, 2024, doi:10.3390/ijns11010003_

Round 1

Reviewer 1 Report

Comments and Suggestions for Authors

Thank you for the opportunity to review the article “Parent reports of developmental services utilization after newborn screening.”  This is important work, especially as the newborn screening field is focusing more on long-term follow-up after a positive newborn screen.  I think your discussion is excellent and really brings home the importance of this work.  However, as a reader, I got lost getting there.  I have made suggestions to reduce the amount of data you share with the reader in hopes of highlighting your points more clearly in the analysis.

Please see the attached Word document.

Author Response

Reviewer 1:

Comment 0: Thank you for the opportunity to review the article “Parent reports of developmental services utilization after newborn screening.” This is important work, especially as the newborn screening field is focusing more on long-term follow-up after a positive newborn screen. I think your discussion is excellent and really brings home the importance of this work. However, as a reader, I got lost getting there. I have made suggestions to reduce the amount of data you share with the reader in hopes of highlighting your points more clearly in the analysis.

  • Response 0: Thank you for your detailed and valuable comments and suggested edits. We believe our changes based on your feedback, specifically to the methods and results, have improved our manuscript. Our responses to each comment are described below, and changes are entered into the new manuscript in red.

Overarching comments:

Comment 1: Throughout the analysis you tell me how many families used all 3 developmental type services which implies more is better. Is it better to use 2 vs 1? What does it tell us if families only use private therapy? What does it tell us if they don’t use private therapies? I would hope families would use Part C and B but I’m not sure it matters if they use private therapies. Help me understand the value of knowing all 3 and if families used more than one type of service.

  • Response 1: Why families use specific services is an important question for future research. On one hand, families may use private therapies because there are fewer eligibility requirements to access services, services can be received at the hospital where they are already accessing other medical services, etc. On the other hand, families may not use private therapies because their insurance does not cover therapies, it is more convenient to receive services in the home with Part C, etc. You raise an important point, however, because we do not want to imply that more services are necessarily better. But rather, we are hoping our results demonstrate the significant need of children and families following a diagnosis after NBS. To clarify this point, we have added a sentence in the discussion that reads: “Finally, our study does not examine the effectiveness of these types of develop-mental services. There is some evidence that EI has additive benefits for children with complex medical diagnoses. Relatedly, our study does not examine the additive benefit of multiple types of services; it is unknown, for example, whether accessing both EI and private therapies is beneficial” (page 15, line 136).

Comment 2: There is a lot of descriptive data here and I worry the point is getting lost. Maybe for the exploratory you can just focus on utilization vs all the other information (who referred, when, etc.) since these children are included in the broader analysis already presented.

  • Response 2: Based on this comment, and also additional comments below, we have significantly revised our results section to reduce the amount of descriptive data in the text. We point readers to find additional data in Tables 5-7 (See 3.2.1 and 3.2.2).

Specific comments:

Comment 3: Line 35, page 1 you say, “diagnosed with an NBS condition”. You should specify here you mean diagnosed as a result of a positive NBS. This is clear later but not in the lit review portion. Line 115, page 24 you do the same thing. I think it is diagnosed due to a positive newborn screen which to me implies they then got medical care early in life.

  • Response 3: Thank you for this suggestion. We have revised the first sentence to read, “The extent to which children diagnosed as a result of NBS use developmental services is unknown due…” (page 1, line 35) and the second sentence to read, “First, while our findings suggest that many children diagnosed as a result of NBS ultimately…” (page 15, line 112).

Comment 4: Lines 51-54, page 2. You say, “Of 51 the 34 conditions examined, most (n = 29) conditions put children at a moderate to high 52 risk of developmental delay and/or high risk of medical complexity, with cascading negative developmental outcomes [15].” Is this true even with early treatment? You need to say that if yes. All conditions on the RUSP should have a high risk of developmental delay or medical complexity or death without treatment.

  • Response 4: Thank you for identifying this missing qualifier. Yes, this is true even with early treatment. We have revised the sentence to read, “Of the 34 conditions examined, most (n = 29) conditions put children at a moderate to high risk of developmental delay and/or high risk of medical complexity, with cascading negative developmental outcomes, even after early detection and treatment” (page 2, line 53).

Comment 5: I love table 2. Great info to justify the 3 conditions you choose.

  • Response 5: Thank you for this feedback. In response to Reviewer 2, we made a minor update to this table to included treatment options for CH and Pompe disease.

Comment 6: Are the parents from all over the US? World? Just NC? If you gathered where families were currently living, please include that detail in section 2.1.

  • Response 6: Our sample is all US-based children and their families, given our focus on Part B and Part C. Our third inclusion criteria reads, “Children primarily lived in the United States for the first 3 years of life” (page 7, line 28). In addition, Table 3 provides further demographic breakdown of geographic region (South, Midwest, Northeast, West) (page 8).

Comment 7: Line 47, pg 7. You said, “we reached out to specialty clinics and NBS follow-up centers”. Is this in the US? It seems like it was only the US given the demographic info you report.

  • Response 7: Yes, this is in the US. We have revised this sentence to read, “we reached out to US-based specialty clinics and state NBS follow-up centers to share our survey with specific patient populations” (page 7, line 49).

Comment 8: Which disorders were included is a little confusing. It looks like you included all disorders but then focused on SCID, CH, and Pompe for another analysis. Is this correct? If so, under section that you included all disorders in your analysis of the RQs. You do call out the exploratory analysis, but I missed that you were looking at all disorders as well.

  • Response 8: Thank you for letting us know this wasn’t clear. You are correct in that we first conducted analyses will include all survey responses, and then a subsequent focus on SCID, CH, and Pompe. We have revised the sentence to now read, “Using data from an online survey of parents representing children with a range of NBS conditions, the current study addresses three research questions…” (page 6, line 11).

Comment 9: Section 3.1, page 7. Why do you think you got mostly white? Was your survey only in English? If not, tell us what languages were available.

  • Response 9: The survey was only available in English. We have added this information on page 7, line 39, now reading “The survey was available in English only.” Consistent with previous registry/survey studies of families of children with rare diseases, our convenience sample was mostly white and potentially not representative, which is a limitation we acknowledge on page 15, line 117.

Comment 10: Line 55, pg 7. The children were 5.7 years at the time of the survey. Please specify their age at the time of the survey.

  • We have revised this sentence to read, “On average, children were 5.7 years old at time of survey completion…” page 8, line 58.

Comment 11: Line 62, page 8. What did you do with the 3 disorders not on the RUSP? It looks like you included them. Defend why they were kept in for analysis given your premise or throw them out and redo the analysis. Also, were you able to control for whether a state screened for the disorder the child had at the time of birth?

  • Response 11: We included any child who was diagnosed with a condition as a result of newborn screening. Specifically, our criteria for inclusion in the analyses was children were identified via NBS and formally diagnosed prior to their first birthday (page 7, line 27). All seven parents of the children who were diagnosed with non-RUSP conditions confirmed their child was diagnosed as a result of a NBS screen. Krabbe (n = 5) is now considered a RUSP condition, but it was not a RUSP condition at the time of survey completion. We added this as a note in Table 4.

Comment 12: Pg. 8, Table 3. For age, can you add the unit “yrs”?

  • Response 12: We have modified the table and it now reads “Age in years” (Table 3).

Comment 13: The results in Section 3.2 would be nice to have in a table. I am having a hard time figuring out how many parents used EI (regardless of whether they used other services), Private, and/or Part B. The info on the combinations is clear. It would be good to know if only 61 families used EI or other families who were not referred also used it, etc.

  • Response 13: Based on this comment and your additional comments above, we significantly revised Section 3.2 to be more streamlined, and encourage readers to look at Table 5 - 7 for more descriptive information. In addition to significantly modifying Table 8 (see response to Comment 20 below), we added the “full sample” for easy comparison to the exemplary conditions.

Comment 14: Table 5 is overwhelming. Can you collapse categories? For example, age for referral to EI, you can do an over 18 months group or even an over 12 months group. Another way to cut down on the data is to not report what therapy was received. I think we just need to know if they were referred for a service, if they were found eligible, and if they enrolled. If you keep the therapies, roll them into larger therapy types. You have small n’s and I am not sure it matters if the reader sees that 2 people got “other”.

  • Response 14: Thank you for this suggestion. We have significantly modified Table 5. For example, instead of including data on individuals who said they were not referred (or didn’t know if they were referred) we only report data for number and percentage of individuals who were referred. We also collapsed the options for age of referral (before 12 months, 13-24 months, and 25-36 months). Additionally, we listed only the top three most utilized services, instead of all potential options. For consistency, we did the same in Table 6 and 7.

Comment 15: I am curious if you looked to see if there was a racial/ethnic/primary language/income difference between those who did and did not use EI and Part B. I wonder if we are not reaching certain families. That may be beyond what you focused on, however.

  • Response 15: We agree this is an important research question to consider, although also agree it is outside the focus of this paper. In our limitations, we describe the need for future studies with more systematic sampling to examine whether/how service utilization rates differ based on race, ethnicity, and SES (page 15, line 117).

Comment 16: Lines 102-104, page 12. Is your font too small? I think the font changed.

  • Response 16: Thank you for pointing that out. All font size should now be consistent, size 10.

Comment 17: For private therapies, did those families differ from those who did not go that route? (region, income, race, disorder, etc.)?

  • Response 17: Comparisons like this was outside the scope of this paper, but is an important avenue for future research once more systematically collected data are available.

Comment 18: For 3.2.3 – How many of the children who did Part B also did Part C? I think it is helpful to know if someone came to the system later.

  • Response 18: About 9% (n = 14) of our sample used both Part B and also Part C, and additional 12% (n = 19) used all three services (page 9, line 81). Very few children used school-based services only (1.3%; n = 2), suggesting that services were initiated and received relatively early for those who needed them.

Comment 19: For the Exploratory analysis, help me understand why I care about private therapy usage. I think it is more interesting to understand the use of public programs like Part C and Part B.

  • Response 19: Our overall goal was to capture usage of all developmental services because we think this in turn reflects the existence of developmental delays. For example, it is possible that a child has significant delays, but parent preference resulted in only using private therapies. We wanted to capture this in our results. For instance, more children (n = 4) with CH used private therapies only when compared to EI-only (n = 1) and when compared to EI and private (n = 3). Future studies are needed to understand when/why children use one service instead/or in addition to the other.

Comment 20: Table 8 is onerous. I don’t think I need to see each child. Consider doing the table on condition type and how were referred to EI, found eligible.

  • Response 20: Thank you for this comment. Based on this comment and comments from Reviewer 2, we significantly updated Table 8. It is now a summary table, indicating the rates of overall use, rates of using EI, private therapies and school-based services.

Comment 21: For the Exploratory Analyses (section 3.2.4) you stopped putting in percentages. Why is that?

  • Response 21: Thank you for pointing out this error. We have added in the percentages in section 3.2.4.

Comment 22: You have listed some key limitations. Another one to consider is that you had higher engagement with developmental services because you have mostly white English speakers and/or those who are connected to the groups you recruited through are connected to the developmental program system.

  • Response 22: We appreciate this comment, and agree this is an important limitation to include. We have added this on page 15, line 123, which now reads, “Also worth noting, it is possible we had higher engagement with developmental services because our sample was mostly White and relatively well-connected in their disease space given our recruitment methods. These families potentially were also more connected to the developmental service programs available to them.”

Reviewer 2 Report

Comments and Suggestions for Authors

This manuscript reports a survey of parents/guardians of children with conditions diagnosed through newborn screening with respect to their use of several types of developmental services.  The topic is timely, the research appears well-conducted, and the results are valuable.  I have only a few relatively minor suggestions for the authors.

1) Page 2, lines 45-46: "... growing evidence demonstrates that not all symptoms for many disorders are resolved after early detection and initial medical treatment."  This is an understatement. It has been well-known for many decades that early detection and treatment are only partially effective for virtually all conditions on the RUSP.  The manuscript language seems to suggest this is a newer revelation.  Similarly, the line language on lines 54-56 on page 23 suggests we are just learning this feature of these conditions.  Obviously, we need to learn much more, to which this research contributes, but the knowledge about partial efficacy has been long-standing.  This language might be tempered accordingly.

2) I did not find the tables with the child-specific data to be worth the space in the manuscript.  Summary data may be sufficient with, potentially, an addendum or accessible additional data online for readers who are interested in that level of information.

3) Table 2 on page 6 - 7 should include the major treatment modalities for each condition.  This is included for SCID but not the other conditions.

4) I was hoping the Discussion section would put greater emphasis on the need for a more comprehensive national approach to data collection on long-term outcomes for affected children.  The type of study here is very valuable, but only because we lack a system for collecting such data.  Further, a more comprehensive system for follow-up allows comparisons of treatment modalities to assess their efficacy.  If the authors feel comfortable, I think some stronger advocacy here would be appropriate.

Author Response

Reviewer 2:

Comment 0: This manuscript reports a survey of parents/guardians of children with conditions diagnosed through newborn screening with respect to their use of several types of developmental services.  The topic is timely, the research appears well-conducted, and the results are valuable.  I have only a few relatively minor suggestions for the authors.

  • Response 0: Thank you for your feedback. We have made modifications based on your comments, and believe our manuscript is improved. Our responses to each comment are described below, and changes are entered into the new manuscript in red.

Comment 1: Page 2, lines 45-46: "... growing evidence demonstrates that not all symptoms for many disorders are resolved after early detection and initial medical treatment."  This is an understatement. It has been well-known for many decades that early detection and treatment are only partially effective for virtually all conditions on the RUSP.  The manuscript language seems to suggest this is a newer revelation.  Similarly, the line language on lines 54-56 on page 23 suggests we are just learning this feature of these conditions.  Obviously, we need to learn much more, to which this research contributes, but the knowledge about partial efficacy has been long-standing.  This language might be tempered accordingly.

  • Response 1: Thank you for pointing this out, and we agree with your comment. We have tempered the language and updated the sentence to read, “…it is known that not all symptoms for many disorders are resolved after early detection and initial medical treatment…” (page 2, line 45).

Comment 2: I did not find the tables with the child-specific data to be worth the space in the manuscript.  Summary data may be sufficient with, potentially, an addendum or accessible additional data online for readers who are interested in that level of information.

  • Response 2: Thank you for this comment. Based on this comment and comments from Reviewer 1, we significantly updated Table 8. It is now a summary table, indicating the rates of overall use, rates of using EI, private therapies and school-based services.

Comment 3:  Table 2 on page 6 - 7 should include the major treatment modalities for each condition.  This is included for SCID but not the other conditions.

  • Response 3: Thank you for pointing out the inconsistency in Table 2. We have added the treatment for CH (“Treatment includes hormone replacement medication”). We also added the treatment for Pompe (“Treatment includes enzyme replacement therapy”).

Comment 4: I was hoping the Discussion section would put greater emphasis on the need for a more comprehensive national approach to data collection on long-term outcomes for affected children.  The type of study here is very valuable, but only because we lack a system for collecting such data.  Further, a more comprehensive system for follow-up allows comparisons of treatment modalities to assess their efficacy.  If the authors feel comfortable, I think some stronger advocacy here would be appropriate.

  • Response 4: We agree this is an important point to make. We have added the following sentence to the discussion, which now reads, “Our study demonstrates how parent-reported outcomes can generate initial data and findings but also highlights the need for more systematic data collection as part of NBS follow-up. Indeed, we hope this type of small-scale study helps justify a comprehensive national approach to data collection” (page 13, line 45)